# Danger Zone or Newfound Freedoms: Exploring Women and Girls’ Experiences in the Virtual Space during COVID-19 in Iraq

**DOI:** 10.3390/ijerph20043400

**Published:** 2023-02-15

**Authors:** Najat Qushua, Alli Gillespie, Dechol Ramazan, Sunita Joergensen, Dorcas Erskine, Catherine Poulton, Lindsay Stark, Ilana Seff

**Affiliations:** 1Brown School of Social Work, Washington University in St. Louis, St. Louis, MO 63130, USA; 2UNICEF, New York, NY 10017, USA

**Keywords:** online violence, gender-based violence, virtual services, COVID-19, Iraq, social media, virtual space

## Abstract

During the COVID-19 pandemic, women and girls across the globe faced increased reliance on the digital space to access education, social support, and health and gender-based violence (GBV) services. While research from the last three years has explored how women and girls navigated and responded to their new virtual reality, minimal evidence has been generated from low-resource settings where access to technology may be limited. Further, no studies to date have examined these dynamics in Iraq, where women and girls already face numerous threats to safety due to various forms of structural violence and patriarchal family structures. This qualitative study aimed to examine women and girls’ experiences in the digital space during COVID-19 in Iraq, including the benefits and risks of engagement as well as how access to the digital space was controlled. Data for the present analysis come from the authors’ larger multi-country study investigating women and girls’ safety and access to GBV services in the context of the COVID-19 pandemic and related public health measures employed to control the spread of the virus. In Iraq, semi-structured key informant interviews were conducted virtually with fifteen GBV service providers. Following the translation and transcription of interviews, the thematic analysis highlighted several benefits and challenges women and girls experienced as they tried to access and utilize technology for schooling, support services, and obtaining and spreading information. While many women and girls increasingly and successfully relied on social media to spread awareness of GBV cases, key informants noted that women and girls also faced increased risks of experiencing electronic blackmail. In addition to a substantial digital divide in this context—which manifested as differential access to technology by gender, rural/urban status, and socioeconomic status—intrahousehold control of girls’ access to and use of technology left many adolescent girls unable to continue schooling and contributed to their further marginalization and consequent decline in well-being. Implications for women’s safety and mitigation strategies are also discussed.

## 1. Background

The SARS-CoV-2 outbreak of 2019 (COVID-19) has affected the world’s population and impacted every aspect of humans’ daily lives. Beyond the numerous challenges, the pandemic has brought to health systems globally and in the Middle East and North Africa (MENA) region, it has led to substantial economic and social consequences for women and girls [1,2]. In Iraq, the COVID-19 pandemic intersected with pre-existing gender inequalities, war trauma, political instability, and protracted humanitarian, sectarianism, and socioeconomic crises to further exacerbate discrimination and inequalities among society’s most marginalized and vulnerable groups [3,4,5,6].

Although the marginalization and victimization of women and girls in Iraq predate the United States-led invasion in 2003, Iraqi women have faced additional and heightened threats to their well-being, security, human rights, and education in the period after [5,7,8]. For example, research finds that, since 2003, women have faced an increase in gender-based violence (GBV), including rape, honor killings, domestic violence, and kidnapping [5]. A mapping report prepared by three advocacy and aid organizations in Iraq indicates that between 2003 and 2018, women and girls suffered human trafficking, forced marriage to ISIS fighters, female genital mutilation (FGM), child marriage, and virginity testing, among other forms of sexual and GBV; these incidents were carried out in both public and private spaces and perpetrated by security forces and other official entities as well as unofficial armed groups [9].

Social and gender inequalities during pre-pandemic times have been further exacerbated since the outbreak of COVID-19, increasing women and girls’ vulnerabilities in a range of sectors [10]. In an effort to prioritize investments in and access to essential health services [11] while simultaneously limiting movement in order to contain the spread of the virus, women faced heightened restrictions on their agency. Closures within the GBV prevention and response system, including reduced legal protection and access to social and health care services, resulted in worsening conditions for women and limited their chances to seek appropriate support. Recent research finds that women were forced to stay in abusive marriages and lost custody or financial support during this period [12,13]. Moreover, the lack of state and formal legal services revived prior tribal laws and pushed individuals to seek justice in tribal courts where women’s rights are consistently denied [13].

As was the case during the pandemic across the globe, GBV service providers, including civil society and advocacy organizations, made a complete or partial shift to providing remote services using landlines, cell phones, digital spaces, and social media platforms. Studies conducted in other settings suggest that the utilization of virtual services and remote communication tools allowed for the continuity of service provision during lockdown periods, movement restrictions, and facility closures [14,15]. Moreover, the digital model of service provisions can improve service efficiency, coordination, and linkages and be offered easily by non-professionals, presenting itself as a low-cost sustainable model [15]. However, studies from this period also highlight the exclusion of marginalized and vulnerable groups of women and girls who face multiple barriers to accessing and using digital platforms and, therefore, may not benefit from virtual services. Among these barriers are limited or no access to the Internet, lack of resources or skills required to engage in virtual services, challenges in finding a safe time and space to connect with services remotely, and a preference by some women to communicate through face-to-face interactions [14,16,17]. 

Previous research on women and girls’ engagement in social media and other online spaces in Arab nations highlights both benefits and risks to use. Most of the existing literature on Arab women’s use of social media, specifically, relates to women’s participation in the Arab Spring uprising of 2011 [18,19]. During the Arab Spring, women used social media and online platforms to voice their plight and transmit empowering messages. Findings show that online platforms allowed marginalized groups in societies, such as women, to enhance their political activism and overcome barriers to women’s participation in activities in the public sphere. Other studies from the region argue that social media benefits some women in Arab Muslim societies by providing them with vital access to information and social networks in ways that circumvent societal boundaries, serving as a medium of self-expression, fostering newfound freedoms, and empowering users [20,21,22]. 

However, as posited by the Cyber Feminist theory, the virtual space may hold a dual influence over women and girls’ lives. The theory argues that although women can enjoy agency and control online in transcending their gendered experiences and sexual realities, the online world may hold limited power to impact offline realities, even for those women who feel liberated in the virtual world [23]. For example, research suggests that the majority of women in the Arab world may engage in self-censorship online in adherence to social norms. While some studies describe self-censorship as an act of agency within sociocultural boundaries, the behavior remains restrictive. Other studies report women’s use of pseudonyms to avoid surveillance, and Al-Maimani [20] concluded that empowerment gained in the virtual space is not often reflected in women’s offline world wherein surveillance and control by male relatives continue and further confine women’s freedom online. Furthermore, women and girls may experience violence in the digital space. Online violence is merely an extension of violence faced offline and can include discrimination, online harassment, doxing, sexual abuse, and sharing sexual and private images without consent [24,25]. 

However, to date, no studies have explored these phenomena during the COVID-19 pandemic in Iraq, when GBV services, schooling, social interaction, and a host of other activities rapidly shifted to the digital sphere. Data collected during this period may offer unique insight regarding the risks of online engagement as well as how the digital divide may lead to inequitable experiences and outcomes for women and girls within the same setting. Employing qualitative insights from GBV service providers, this paper seeks to answer two primary research questions: (1) What were the benefits and risks for women and girls who accessed and engaged in the digital space during COVID-19 in Iraq? (2) How was women and girls’ access to the digital space compromised or controlled during COVID-19 in Iraq? 

## 2. Research Design

The data analysis procedures described here are part of a larger, mixed-methods research study that was conducted in four countries: Iraq, Italy, Guatemala, and Brazil. The aim of the larger study was to understand women and girls’ experiences of safety and well-being as well as their access to and availability and quality of GBV services [26,27,28]. The study also sought to understand service adaptations made by providers to promote continuity of service in the face of public health control measures. This paper presents an analysis using the Iraq data only.

### 2.1. Setting

Iraq, officially the Republic of Iraq, is a country located in the Middle East, bordering Turkey, Iran, Syria, Jordan, Kuwait, and Saudi Arabia. The population is estimated at approximately 40 million (July 2021), with 75–80% of the population identifying as ethnically Arab and 15–20% as Kurdish. The two official languages in the country are Arabic and Kurdish and the official religion is Islam [29]. Approximately 49% and 60% of the population are female and under 24 years of age, respectively [30].

The country has suffered for decades from war, political conflicts, sanctions, and invasion, resulting in a large number of internally displaced persons (IDPs) [5]. In addition, Iraq hosts many Syrian refugees [12]. Social and gender inequalities pre-existed COVID-19 in Iraq, but the crisis has further intensified political, protracted humanitarian, security, and economic challenges. COVID-19 has also contributed to a deterioration in socioeconomic well-being and reduced people’s access to support and financial resources, which has, in turn, deepened the gender inequality gap [31]. Most of the social services and charities in the country are provided by local and international NGOs that rely mainly on external and international funding [12,32].

### 2.2. Study Procedures and Tools

The research team worked with UNICEF Iraq to recruit key informants using purposive sampling. Initially, the UNICEF team in Iraq contacted individuals who worked for non-governmental organizations (NGOs), civil society organizations (CSOs), and other relevant service providers by email. Key informants were purposively selected based on the eligibility criteria, which required participants to be at least 18 years old and have professional experience providing GBV services both before and during COVID-19. Eligible key informants worked in a range of related sectors, including human rights, justice, and psychosocial support. In the initial email, the UNICEF team included an information sheet about the study. Invited individuals who agreed to participate, as expressed in a response email, were contacted by a member of the Washington University research team to schedule an interview. Fourteen service providers agreed to participate in the study, and two of these providers suggested the team speak with two additional key informants who were also invited to participate, for a total of 16 key informants. Two participants engaged in a paired interview, and one service provider was accompanied by two colleagues who listened to the conversation but did not participate themselves. As such, in total, fourteen KIIs were completed between 4 May and 31 July 2021. Key informants in the final sample represented a range of service provider types, including project managers, country directors, case managers, and protection focal points, among others. Services provided by key informants and their organizations included legal assistance, mental health and psychosocial support, awareness raising on women’s rights, case management, immediate support services, and empowerment workshops.

Semi-structured key informant interviews were conducted online using the Zoom platform. Prior to the start of each interview, the researcher reviewed the study purpose, confidentiality, and the voluntary nature of participation and allowed participants to ask any questions they had about the study or their participation. All interviewees were asked to provide verbal consent to participate in the study and to have the discussion recorded for transcription purposes. Eleven interviews were conducted in Arabic and three were conducted in English; interviews lasted approximately 45 min to one and a half hours. 

All participants provided verbal consent to participate and to have the conversation recorded before the discussion began. Ethical approval for all study procedures was granted by the Health Media Lab’s Institutional Review Board (HML IRB Review #361GLOB21).

### 2.3. Tool Development

The KII and FGD interview guides were developed in collaboration with UNICEF team members in Iraq, during which some of the questions were removed and modified following the Iraqi team’s recommendations. The semi-structured interview guides included questions and probes related to changes in the availability of and access to GBV services during the pandemic, how different dimensions of identity compromised women and girls’ access to services, women and girls’ experience of violence, exploitation, and early marriage, the ways in which services shifted to a virtual format, challenges faced by service providers during this period, and others. The interview guides were translated into Arabic by a member of the research team, and then a member of the Iraqi UNICEF team provided a Kurdish translation. The Iraqi team approved all translated versions and confirmed the cultural acceptability of all questions. Interview questions were informed by study objectives and focused on gathering information about the impact of COVID-19 on the availability and accessibility of GBV services, challenges key informants faced in providing GBV services during COVID-19, shifting services to a virtual format, women and girls’ safety, and changes in early marriage and sexual exploitation, among others. 

### 2.4. Data Analysis

All audio-recorded interviews were saved, transcribed, and translated into English. Identifying information, such as informants’ names, was removed to maintain participants’ confidentiality and anonymity.

Data analysis of the resulting transcripts was guided by Braun and Clarke’s [33] recommendations for thematic analysis and employed both deductive and inductive analytical approaches. As the broader research question for the larger, multi-site study was the same across sites, we modeled our first draft of the codebook after the codebook used in Italy, where analysis had already been conducted. However, after reading the Iraqi transcripts and memoing, a team of three researchers used inductive approaches to substantially revise the codebook, including removal, addition, and modification of codes. Three coders then coded the text independently, and the codes were compared to enhance inter-rater reliability. After close agreement on the basic themes was achieved, the team finalized the codebook and coded the entire dataset. Transcripts were coded and organized using Dedoose, a web-based program for data organization and analysis. 

Next, the team gathered excerpts relevant to each code and reviewed excerpts for emerging themes. Emerging themes included the following: (1) women and girls’ voices in social media; (2) virtual space as a danger zone; (3) the digital divide, or the unequal access and utilization of digital platforms that exclude certain groups of Iraqi women from enjoying online GBV services; (4) implications of digital divides and virtual services for girls’ education and early marriage; (5) service adaptations and challenges faced by service providers; and (6) mitigating strategies (see Table 1). These themes were then organized as they related to the two primary research questions. A potential source of analytical bias was related to the first author being the interviewer, who pointed at digital space as a unique theme in Iraqi data. To guard against this bias, the research team employed a group approach [34]; an additional two skilled researchers assessed the transcripts independently, and a comparison with the first author’s assessment was conducted. This process resulted in an agreement around the centrality of the theme. 

## 3. Results

In shifting service provision to virtual modalities, service providers were able to better understand women and girls’ complex experiences in the virtual space. According to participants, remote communication, digital tools, and social media opened the virtual world for women and girls to engage and communicate with service providers, family, peers, and the public. However, key informants noted that not all Iraqi women enjoyed this technological advancement and that many were left behind in this shift to the virtual space. Key findings from the data serve to elucidate (1) the benefits and risks for women and girls who had access to and engaged in the digital space and (2) the digital divide and the different ways in which women and girls’ access to the digital space was compromised or controlled across a range of identity dimensions. Challenges in service provision and mitigation strategies employed by service providers are also discussed throughout the results.

### 3.1. Benefits and Risks for Women and Girls Who Had Access to and Engaged in the Digital Space

#### 3.1.1. Benefits: Women and Girls’ Voices in Social Media

Many key informants noted that posting to social media allowed women and girls to share their stories of violence with the public and seek support. Women and girls reportedly narrated first-hand experiences of violence and stories they heard or witnessed happening to other women in their environment. Although the effects of posting did not often carry immediate assistance to the survivors or the narrators, interviewees pointed to positive developments following such posts. A male human rights activist identified as a member of an organization that monitors human rights situations across Iraq described the following: “Most likely, I have seen cases of domestic violence and problems in dark rooms where social media played an important role in the detection, as well as in strengthening the victim and in facing the perpetrator” (1).

The activist went on to describe how social media posts contributed to active conversations about violence against women, giving an example of a woman who died by suicide after experiencing domestic violence: “Her sister spoke via social media and recorded a video, and after the video, the case became a public issue” (1). Other providers observed similar developments; a female lawyer that works in a local organization concerned with legal advocacy for women and girls commented:


*For women who use social media, this change has had a significant impact as these topics have not previously been posted on social media. In the past, we have only raised these topics in schools and universities through our meetings with students, but now we have seen that we have been able to reach a segment of women who have not spoken on such topics but are now speaking through social media (15).*


Participants also noted that women and organizations used social media platforms to encourage women to report incidents of violence. Key informants spoke to how violence against women and girls did not receive appropriate attention among Iraqis during the pandemic, especially as people’s focus shifted to COVID-19-specific health concerns. While participants expressed concern about the implication of this neglect, such as the lack of shelters for women survivors and the lack of legal protection, which in turn discouraged victims from reporting, they observed that social media allowed to counteract and report, as explained by a lawyer who works in legal advocacy services for women: “Posting cases of violence on social media helped and encouraged women to report on domestic violence cases” (15). 

Women broke the silence around violence, and online discussions allowed them to express their suffering: “If a girl burned herself in a governorate, you would find her picture and her story published online. Many cases are discussed online now, especially the ones that are published on public Facebook groups; violence is no longer a hushed subject” (2), a female feminist activist who has worked for many years in organizations that are interested in women’s issues from the north to the south of Iraq noted. Key informants felt that sharing stories via social media strengthened solidarity among survivors of GBV and attracted the attention of governmental actors and human rights organizations who acted to find solutions and provide support for the survivors.

#### 3.1.2. Risks: The Virtual Space as a Danger Zone

**Children’s exposure to online violence.** Most key informants pointed to the various ways in which social media presented risks for women and girls during the pandemic. Service providers reflected on instances where women and children described violent interactions with others online as well as exposure to abusive and harmful content. While families and CSOs invested in digital devices for learning and communication purposes at the start of the pandemic, key informants noted that many did not prepare for or expect the harmful impact on children. As a program manager in a non-profit organization serving people affected by poverty, social injustice, and natural disasters and a child protection worker described, “So what was done is that in one of the NGOs, there was a distribution of tablets to children, but there was no age category, there were no regulations” (10,11). The child protection worker described cases of children being harassed and threatened online: 


*There were a lot of cases of safeguarding where children were being harassed, were being, let’s say, used, were being threatened. So, this was the frustrating fact about exposing children to social platforms and cyberbullying. In one of the cases, there was a, okay, I’m sorry to say this, but there was a very high-risk case of exposing children to pornography, even (10,11).*


**Electronic blackmail**. Many providers discussed a phenomenon they called “electronic blackmail”, which raised significant concern over women and girls’ safety in the virtual space. Women in these spaces encountered different kinds of violence, including sexual and economic abuse. A female lawyer who worked directly with blackmailed women who approached her relief and development organization (specializing in women, peace, and partnership in Diyala province) described the situation in detail.

We noticed that in our area, electronic blackmail has increased. Electronic blackmail is common in our society and has increased since COVID started. This electronic blackmail is often done by extended family members because they often know the family’s circumstances. Electronic blackmail is very common, and I think there needs to be more awareness of it so that women learn to protect themselves. I heard about a situation where the blackmail got so bad that it led to the young women committing suicide. When blackmailed, the young women find it hard to confide in anyone about it (8).

More than half of the participants described electronic blackmail and cyberbullying as an emerging form of violence that required new actions to address its negative impact on women and girls’ everyday lives, health, and mental health. As noted by one female lawyer who works in women’s human rights and violence against women through the Women Leadership Institute in Iraq, “Yes, it increased, especially during and after the pandemic, because the use of social media platforms increased” (13). According to participants’ observations, blackmailers were often males who threatened to post photos of women in socially inappropriate or prohibited bodily positions unless they paid a certain sum of money, thus risking hurt to their reputation and honor and putting them in a vulnerable financial position. Another service provider, discussing the increase in e-blackmail against adolescent girls, explained that blackmailers “send things to the girls like, ‘We are viruses’ or ‘we will destroy the device if you don’t send the picture’. So they end up sending the images, not knowing what to do” (6). Participants believed that due to plenty of free time, financial distress, and job loss, the men who e-blackmailed women and girls often knew them and even had a relationship with them, including romantic or family relationships. As communication technologies and social media became widely available and utilized by women and children during the pandemic, CSOs in Iraq actively sought ways to protect vulnerable and marginalized people participating in virtual activities. Service providers expressed concern that women and children in digital spaces might encounter different forms of violence, particularly during the pandemic when the pace of technology use evolved far faster than they imagined. 

In response to these concerns, CSOs held awareness workshops regarding potential risks on social media and how women and families could protect their loved ones from harm. Mothers shared their fears about (and actual cases of) electronic abuse of their children. A project manager (GBV project with activities for women, men, and children) and staff supervisor shared her organization’s strategy to address these risks:


*Among many things you can do is how we can protect women and girls electronically. One of the awareness workshops we offer is “How to protect me from electronic abuse.” I think they need it now. During the COVID period, everyone had Internet, so the focus was on mobile. They were always busy with their mobile. They had nowhere to go (6).*


A CSO focused on legal aid provided concrete support to women who had been blackmailed, including contacting webpages that specialized in assisting e-blackmail victims in closing the blackmailers’ sites. Providers stressed that collaboration between CSOs and experts in the field of electronic blackmail is needed to fight against these crimes and actions.

### 3.2. The Digital Divide

When service providers were asked how increased reliance on the virtual space during the pandemic impacted the availability and accessibility of GBV services, almost all participants referenced the segment of Iraqi women who were unable to access services online. Analysis of the data revealed multiple and intersecting factors that limited women and girls’ access to the digital space, as noted here by the chair of a relief and development organization in a conflict zone: 


*“*
*There are different groups of women in Iraq. The first one is the group of women who own smart devices and use the Internet. Another group is the one who has no financial means to own smart devices. Lastly, the group that is prohibited from owning a smart device or using the Internet without supervision (8).”*


One key informant articulated how the digital divide impacted service providers’ provision of, and survivors’ access to, online services during the pandemic, noting:


*We were hoping that remote sessions could be held. However, the women from our database that were responsible for helping us invite other women from their village reported that not all women in the village had smartphones or Internet, which prevented us from holding remote sessions. We had also hoped to launch a campaign about COVID through WhatsApp; however, we found that only about 50 percent of the women we know had access to the Internet or have a smartphone (8).*


Another key informant shared a similar sentiment about those who were left behind: “There were areas where our mobile teams couldn’t reach, we used to contact beneficiaries on the phone, and if they didn’t have a phone or a laptop, we couldn’t reach them (2).” 

In addition to financial barriers to accessing technology, many participants shared that women’s, and especially girls’, access to the digital space was often controlled within the household. Key informants explained that some families prevented their female members from accessing or using digital tools, while others allowed access with male or parent supervision only (usually the father or the older brother). A male administrator for child and women protection programs in a CSO elaborated: 

Women with special needs have difficulty getting out of their houses, and it is not easy to meet them, and the parents of the young girls don’t allow going to the centers, and many families are very poor they don’t have mobile phones so the social worker can’t communicate with the girls remotely, and we also have the women who are abused and controlled by their husbands (3).

Some participants felt that men’s reason for regulating women and girls’ use of social media and digital tools was to “protect” them from violating social customs and rules and the associated risks. Beyond families’ desires to protect their young daughters from the danger they might encounter while participating in online sessions, several participants mentioned that some sections of society felt threatened by the knowledge women gained about their rights through communication with feminist and aid organizations. As such, key informants felt that these swaths of society attempted to sever such communication by denying women their right to access and use communication tools, as evident in the following:


*Because we live in a masculine society, everybody thinks that if women have legal knowledge about their rights, they would be a threat to their families, and who teaches them the information about their rights? The organizations, the female social workers, and the female lawyers, the men consider the organizations or the social workers as a threat, and so the men threaten their women, and so the women cut their communication with the organization because it is the source of the information (2).*


In addition to social and familial supervision of women’s access and use of digital platforms, some women were closely watched by their abusers, as shared by one female project manager in a local aid organization in the Kurdistan region in Iraq:


*Now we work in Baa’j (a town in Iraq), in this town even women from well-off families do not have cellphones. When we call them, the mother-in-law or the husband answers. We asked women to come to our centers in Dohuk for training they refused and told us that their husbands who are their guardians do not approve of it. So yes, there are women who have no access to phones (9).*


Another key informant described how some husbands reacted to remote communication between their wives and GBV services: 


*If we are to give services to a woman, the person who caused the violence might be sitting next to her, and we faced many cases like that where as soon as we end the call, the husbands would call us and say that they don’t want us to communicate with their wives, services are incomplete because the feeling of safety is lacking for the women, how would the woman talk about her problems if the person who caused the violence is sitting next to her?! (3).*


According to providers’ accounts, the sociopolitical and socioeconomic status of women and girls influenced their ability to access and use technological tools. Many of the women in this category were described as poor and located far from city centers in rural, conflict, or refugee and IDP areas. Economic barriers to accessing technology were discussed by most participants, as one participant who works closely with refugees within a child protection organization that works in different governates in the country noted, “Then there is the problem of the money. So, for very vulnerable people, uh, it may be to charge for the phone or to charge with the Internet is like a choice and is like not something that they can do continuously” (5).

Providers struggled to support women in these communities. Mobile teams were required to show authorities’ approval for their movement, and in many cases, there was a need for more than one approval since different political forces were present in the area at the same time. As articulated by one key informant: 


*The segment that can’t access the services is the women who live far away, and the poorest of them, and the women who don’t know how to use social media platforms. I didn’t allow my staff to go to these places for their safety, not just because of the pandemic, but as you know, the security situation in Iraq is complicated (13).*


#### Implications of Digital Divides and Virtual Services for Women’s Education and Early Marriage

Participants discussed cases where girls could not continue their education when in-person learning activities switched to an online modality. Remote learning required families to have digital devices and Internet connections available for their children to engage in the learning process, including testing. Women and girls found themselves unequipped with the necessary digital resources to complete their schoolwork. “There are a lot of women who have had difficulty taking electronic exams. These women have borrowed their father’s or brother’s phone to take the exam. This is a big problem, that we didn’t think to this day some women don’t have a phone or are not allowed to have a phone” (15). While some were lucky to have their male family members’ support, others were prohibited from using digital tools for learning, as noted previously, and ended up dropping out of school or being unable to meet institutional requirements, a reality that a lead supervisor of program activities related to GBV in a feminist organization observed:


*In terms of the social situation, there is an impact on education where the quality of education for women decreased, and that is because universities and schools started using distance learning and shifted to online. Universities and schools obliged students to open the camera to be sure of their identity. Many fathers refused to allow their daughters to open the camera, which caused many girls to fail as they could not complete the exams as required, the refusal of the fathers due to the customs and traditions prevailing in society (14).*


Indirectly, the interruption in schooling and learning contributed to an increase in early marriage. Many participants described how before the pandemic, young women and girls used schooling and their desire to acquire education as an excuse to avoid early marriage; however, key informants felt that the transition to online learning, and the consequent inability to continue education for many girls, resulted in many families marrying off their young daughters against their will. One participant who was asked about changes in early marriage since the start of the pandemic answered:


*During this period, there were no schools. Schools were online. Of course, the online school failed because not everyone has a laptop they can use, not everyone has a communication tool to communicate with people. So it increases, a large percentage of early marriage increased during this period, I think more than the period before (6).*


Another participant made a clear connection between online education and the increase in early marriage, noting, “They refused to marry off their daughters because they had to go to school, but now girls don’t go to school, they are home taught via online schools, so electronic education contributed to the increase in numbers of early marriage cases” (2).

However, a few participants discussed the positive impact of the digitization of schooling, highlighting how online education offered more educational opportunities for women and girls in marginalized, vulnerable communities—for example, women in conflict areas who suffered under ISIS rule. A lawyer in a conflict zone shared:


*Despite the increase in responsibilities, I noticed that there is a high percentage of women who decided to pursue a college degree or higher. For example, one of the universities in Lebanon offered an online master’s class. Most of the students who signed up were women from … provinces that are conflict zone that was under the ruling of ISIS (8).*


Another participant believed that online learning might appeal to families living in refugee camps as the virtual modality protected their daughters from the harassment they faced on the dangerous roads to school:


*But, like, at least in Iraq, the main reason why girls are not accessing education apart from, like, is because the family has care because the women are harassed in the street, so they are scared about the road between the home and the school, or because they are worried that something will happen in the school because there were also cases of abuse inside the school, and so on, rather than a belief that women should not be educated. So... maybe it that, like, what was developed during COVID, might be a chance for girls that are old, because like have anything …. the girls are, you know 15, 16 if they go out from the house, something will happen to them and to their honor. They can still be getting access to education opportunities (5).*


Some providers described a similar situation wherein families might refuse to allow their daughters to participate in on-site vocational training while approving their online participation due to fear of harm outside the house.

## 4. Discussion

This study aims to understand the impact of the COVID-19 pandemic on women and girls’ experiences of digital violence and remote service access and availability. Echoing findings from other parts of the world, the results suggest an increase in many types of violence against women and girls in Iraq since the outbreak of the pandemic [35], including e-blackmail, intimidation, and cyberbullying. However, findings also speak to the positive applications of social media during COVID-19, including using platforms to encourage reporting and raise awareness about GBV. Importantly, the present study’s analysis highlighted the lack of or controlled access to the digital space for certain groups of women and girls, which in turn had profoundly negative impacts on their well-being.

Study findings highlight an expansion of social media across Iraq and unpack women’s experiences using online platforms during COVID-19. According to interviewed service providers, social media allowed continuity of communication with beneficiaries, both for direct services and for broader services, including awareness campaigns around GBV and COVID-19. Importantly, however, the expanded availability of technology and virtual services during COVID-19 reached only a portion of Iraqi women, leaving others marginalized and unable to benefit from this expansion. In parallel with studies from other settings [14,16,17,36], our findings affirm these same dynamics in Iraq, demonstrating that poverty, a lack of technology skills (as well as poor or limited electrical and digital infrastructures), and sociocultural norms contribute to reinforcing this marginalization of certain groups of women with respect to digital access, particularly, but not exclusively, in rural areas. Bello-Bravo and colleagues [37] offer one possible redress for this challenge through the utilization of animated videos translated into locally and comfortably spoken languages. These videos are then delivered and shared on the workers’ mobile phones (no need for the women to have their own devices), which have a demonstrated capacity to bypass such barriers, especially in rural and geographically isolated areas [37]. Information delivered and shared in this way, especially in traditionally women’s spaces, would afford one empirically tested and safer option to remediate this problem as compared to continually relying on social media.

Erskine [38] discusses means of reaching women and girls in violent situations without access to phones, due to a lack of device or abuser surveillance. Alternative safe and appropriate means of communication may include integration of GBV services in women-permitted spaces (e.g., food distribution stations), using an alert object or code word to signal abuse in predetermined public spaces, such as pharmacies, markets, and others, and utilization of “no dial or chat” phone options.

While the present study does not directly explore the impact of women’s social media use on gender dynamics within the family, it does draw attention to the scarce empirical literature on the topic [39]. Service providers shared how women and girls subjected to violence shared their personal or other victims’ stories, feelings, and opinions via posting on online social media, bringing hidden stories to the spotlight. Providers noted that online posting served as a mechanism for making private issues public. While this development may lead to “the blurring between private and public [because] it collapses traditional divisions between the home and the street” [39], such transformation of the private into the public also builds awareness among women affected by GBV that they are not alone. Such “solidarity of recognition” [40] is crucial for the redress of systemic trauma generally [41]. While this awareness and solidarity can arise through social media platforms (for example, interviewees discussed positive developments that followed from posting violent stories on social media), any form of safe digital sharing and dissemination of information can suffice [42]. Moreover, from a feminist perspective on the dynamics of communication technologies and violence against women, the increased sharing and diversity of content producers allows various representations of women that affect gender relations in a complex way [43]. More research is needed to assess the strands of gender, sexual, cultural, and racial discourses communicated digitally and their role in affecting culture and norms [43] particularly as it relates to violence against women.

Importantly, digital sharing presupposes digital access. While most literature on the digital divide for women focuses on geographic and socioeconomic barriers to access, we found that, in Iraq, women’s access to the Internet and social media was often controlled by male family members due to fears around women gaining information about their rights, means of coping with or responding to domestic violence, and consequently means for changing familial power dynamics and undermining males’ authority. Our study’s findings also show that some families restricted women and girls’ use of social media and digital devices by either completely prohibiting use or only allowing women and girls to use these platforms with supervision. Some girls and women were forced to drop out of school because they were prohibited from using digital platforms, even for educational purposes. The benefits and consequences of the digital space for women revealed by the data raise the following question: under what conditions are social media and digital platforms empowering, strengthening, or weakening for women survivors of violence, and how do they influence their experiences of violence? More research is needed to address these concerns and to identify information pathways (e.g., in traditionally women’s spaces or religious institutions) that afford delivery without familial or supervisory hindrances.

Lastly, the findings indicate that online violence against women and girls increased in Iraq during the COVID-19 crisis; in particular, the phenomenon of e-blackmail was reported by the majority of key informants as common. While El Asmar [44] uncovered how digital spaces may present an alternative avenue for enhancing women’s participation in public debates, expressing their thoughts, living out their identities, and negotiating power relations, this increased public visibility may also give way to a rise in online violence, especially cyberbullying and a “pandemic of online GBV” [44].

Bali and Omer [45] studied psychological violence against Arab women in social media contexts and found that participants of all ages reported encountering online sexual harassment. According to this study, young women and those with a lower educational level were more likely to experience online abuse. Online violence and e-blackmail harm females’ mental health generally, but especially young women through threats to their families’ honor and reputation [45], and their daughters’ marriage prospects or marriage [46]. Findings from the present study, driven by insights shared by key informants, suggest that organizations should consider holding workshops to equip women and girls with the knowledge needed to protect themselves in the virtual space. However, further research is needed on the implications of online violence for women and girls’ daily lives, including mitigation strategies, mechanisms to support survivors, and a better understanding of the economic and gendered drivers of such violence for male perpetrators.

## 5. Limitations and Strengths

This study has several limitations. First, the study relies on service providers’ descriptions of women and girls’ experiences of violence. Directly accessing and including women and girls’ descriptions in their own voices and through their perspectives in future research would broaden and refine our findings. Second, this study was not initially designed to assess the use of social media and digital platforms; rather, this topic emerged as an unexpected theme during the key informant interviews. Because the study was not designed specifically to detect this central phenomenon, additional research exploring this theme explicitly may better ground these findings or enhance their transferability (i.e., whether it would be useful to use the study findings in another setting). However, given that the interview guides were not focused on this topic, the salience of this theme in the data speaks to the importance of the topic for key informants interviewed in this study. Lastly, the interviews were conducted at a specific point in time, which may not capture the fully diverse experiences of violence among distinguished groups of women in Iraqi society (including internally displaced women, women allegedly affiliated with ISIS, and refugee women). It may also inadvertently highlight certain at-the-time points of worry (e.g., electronic blackmail). Such a limitation can be productive for this very reason, i.e., can capture worries especially prevalent at the time of the research that might otherwise not appear because other issues are more pressing. As such, longitudinal studies exploring girls’ and women’s experiences are crucial to understanding their complex virtual and physical realities.

Strengths of this study include generating rich descriptions, using a culturally adapted interview guide, and conducting interviews in participants’ chosen language. These factors facilitated deep engagement between the interviewer and study participants. Participants also represented different governorates or sectors and worked with a range of urban, rural, refugee, and IDPs from diverse religions (Suni Islam, Shia Islam, Yazidi) and ethnicities, primarily Kurdish and Arab. Participants’ diversity allowed us to capture a broad spectrum of voices, experiences, and perspectives.

## 6. Conclusions

The present study enriches the literature on Iraqi women and girls’ online experiences during the COVID-19 pandemic. Findings allow us to understand this population’s experiences at a time when public health measures were employed to contain the spread of the virus. Understanding the intersecting contextual factors that shape these experiences and exacerbate pre-existing inequalities, discrimination, and risks that women and girls face in their everyday lives may contain implications for services providers in the GBV sector and related services such as health, mental health, and education. Given the findings around gender differences in access to and utilization of technology based on socioeconomic and/or cultural barriers, further exploration of options for programming that circumvent these restrictions and can reach the most isolated and hard-to-reach women and girls is critical. More resources must be allocated to support mobile teams to be able to reach women and girls in remote settings who may not have access to technology. Specifically, GBV actors must continue to explore and test innovative service delivery models that ensure a continuum of care during public health restrictions, such as delivery through mosques, health facilities, and women-dominated spaces. Further, findings stress the need to support women and girls in coping with violence at home, in public, and in virtual spaces. Such solutions must include support for women and girls to navigate safely in the virtual space and mitigate emerging risks such as electronic blackmail. Additionally, policies and programs may usefully work with men and boys to curtail men’s control over women and girls’ access to technology and address men’s stressors as drivers of violence. Lastly, considering the value in sharing personal stories of victimization and violence online, as reflected in the findings, providers might seek to expand such spaces where women and girls can safely share their personal stories with others, receive support, and learn how to access services when experiencing violence.

## Figures and Tables

**Table 1 ijerph-20-03400-t001:** Emerging themes and example quotes.

Theme	Description	Example Quotes
Women and girls’ voices in social media	The ways posting to social media allowed women and girls to share their stories of violence with the public and seek support	“Posting cases of violence on social media helped and encouraged women to report on domestic violence cases” (15). “Most likely, I have seen cases of domestic violence and problems in dark rooms where social media played an important role in the detection, as well as in strengthening the victim and in facing the perpetrator” (1).
Virtual space as a danger zone	The risks social media presented for women and girls during the pandemic	“We noticed that in our area, electronic blackmail has increased. Electronic blackmail is common in our society and has increased since COVID started” (8). “All family members stayed at home for a long time, so girls spent most of their time in the virtual space where they were electronically blackmailed; many cases of girls who were blackmailed caused a lot of trouble to families” (12).
Digital divide	The unequal access to and utilization of digital devices and virtual spaces	“There are different groups of women in Iraq. The first one is the group of women who own smart devices and use the Internet. Another group is the one who needs financial means to own smart devices. Lastly, the groups prohibited from owning a smart device or using the Internet without supervision” (8). “When we held training for teenage girls, and as you know, it is difficult for teenage girls to have their own phones (2).
Implications of digital divides and virtual services for girls’ education and early marriage	The ways the digital divide in the country influenced different aspects of women and girls’ lives during COVID-19	“Universities and schools obliged students to open the camera to be sure of their identity. Many fathers refused to allow their daughters to open the camera, which caused many girls to fail as they could not complete the exams as required. The refusal of the fathers due to the customs and traditions prevailing in society” (14). “They refused to marry off their daughters because they had to go to school, but now girls don’t go to school, they are home taught via online schools, so electronic education contributed to the increase in numbers of early marriage cases” (2)
Service adaptations and challenges faced by service providers	Difficulties service providers faced in adapting to the new reality of the pandemic in order to protect women and girls in their everyday lives	“On the few occasions where we try to do something online because we managed to provide credit, and we were reaching a community with more possibility, it was still not particularly successful because the staff is supposed to have much more preparation and totally change the approach to the modality, the material, the knowledge about how to use technology… And this was not possible because one day to do, then it was okay to stop doing that and start to do online (5).
Mitigating strategies	Strategies employed or suggested by service providers to mitigate risks women and girls face in the virtual space	“We tried to help women, for example, we communicated with a page called the E-Blackmail Warriors page and other pages and they helped women who had been blackmailed by trying to close the blackmailer’s page, and sometimes they were communicating directly with the blackmailer” (15) “One of the awareness we offer is “How to protect myself from electronic abuse” (6)

## Data Availability

Data are not available due to privacy restrictions.

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
