# Peer review of "Danger Zone or Newfound Freedoms: Exploring Women and Girls’ Experiences in the Virtual Space during COVID-19 in Iraq"

_ijerph, 2023, doi:10.3390/ijerph20043400_

Round 1

Reviewer 1 Report

Please, refer to the comments in the attached file.

Author Response

Please see the attached reviewer comment table. 

Reviewer 2 Report

Dear authors

After reviewing your manuscript, i Have some questions:

1.- It is essential that authors complete all the information related to the ethical aspect of their study.

1.1.Nowhere in the document is it detailed whether this study has been approved by any official ethics committee, including whether the participants were informed and, if so, whether they signed or expressly consented to participate.

1. Should define even better than in the case of adult participants, how they have recruited the girls. 

2.- in relation to the coding and analysis of the content of the interviews. They could indicate the main categories and subcategories found in their study. It would be of great interest to know them in order to understand your approach to analysis.

Thanks

Author Response

(The authors gave the same response as above.)

Reviewer 3 Report

Thank you for inviting me to review this article. I found it to be well written, relevant and interesting.  Very minor revisions necessary. 

Author Response

(The authors gave the same response as above.)

Reviewer 4 Report

The paper contains sufficiently new and suitable information, and it adheres to the journal’s standards. The topic and level of formality are appropriate for the journal`s readership. Its style and readability are suitable. There is a huge amount of information given throughout the article, but I would suggest revising the paper. 

The methodological concept is clear, and the selected methodology is scientifically appropriate. But the Tool development is very poor.

I miss recent relevant literature in this area. The literature review is very poor. I suggest citing also TOMAŽIČ, Tina, BESSA VILELA, Noemia. Ongoing criminal activities in cyberspace : from the protection of minors to the deep web. Revija za kriminalistiko in kriminologijo. 2017, 68, 4, 412-423. ISSN 0034-690X.

The authors should use any tables or graphics to show the results. The results are poorly transparent.

Further, I recommend rewriting the conclusion. The concluding remarks should be more specific and better explained.

In summary, the article is sufficiently interesting to warrant publication, but it needs major revision. Please follow all the comments above.

Author Response

(The authors gave the same response as above.)

Round 2

Reviewer 4 Report

I confirm all the changes made.